# Reply to Böning et al. Comment on “Ceruti et al. Temporal Changes in the Oxyhemoglobin Dissociation Curve of Critically Ill COVID-19 Patients. *J. Clin. Med.* 2022, *11*, 788”

**DOI:** 10.3390/jcm11154547

**Published:** 2022-08-04

**Authors:** Samuele Ceruti, Bruno Minotti, Andrea Glotta, Maira Biggiogero, Giovanni Bona, Martino Marzano, Pietro Greco, Marco Spagnoletti, Christian Garzoni, Karim Bendjelid

**Affiliations:** 1Department of Critical Care, Clinica Luganese Moncucco, 6900 Lugano, Switzerland; 2Emergency Department, St. Gallen Cantonal Hospital, 9000 St. Gallen, Switzerland; 3Clinical Research Unit, Clinica Luganese Moncucco, 6900 Lugano, Switzerland; 4Department of Internal Medicine, Clinica Luganese Moncucco, 6900 Lugano, Switzerland; 5Emergency Department, Clinica Luganese Moncucco, 6900 Lugano, Switzerland; 6Intensive Care Division, Geneva University Hospitals, 1205 Geneva, Switzerland

We would like to thank Böning et al. for all the important issues raised in the present commentary [1], giving us the possibility to review all our data and to discuss in more detail this emerging, interesting topic. In line with what was reported by Vogel et al. [2], we found a left shift in Oxygen-Hemoglobin Dissociation (ODC) curve affinity, measured throughout a p50 ODC curve analysis, but only in the first 3 days of the Intensive Care Unit (ICU) stay (p50 early), compared with the last 3 days (p50 late; 26.1 ± 4.7 mmHg, *p* < 0.01 vs. 26.2 mmHg ± 7.0 mmHg, *p* = 0.13) [3].

Regarding our formula, the p50 calculation was performed with the Hill formula; Dash et al. reported, however, an accuracy for the Hill coefficient only between SO_2_ of 30% and 98% [4]. They proposed a variable coefficient depending on pO_2_ for the whole saturation range, which we implemented ($2.82-1.2\times 10^{-\left(\frac{{\mathrm{pO}}_{2}}{29.25}\right)}$) [4]. Finally, p50 was accordingly calculated by this formula:$${\mathrm{pO}}_{2}\times {\left(\frac{1-{\mathrm{SO}}_{2}}{{\mathrm{SO}}_{2}}\right)}^{\left(\frac{1}{2.82-1.2\times 10^{-\left(\frac{{\mathrm{pO}}_{2}}{29.25}\right)}}\right)}$$

Due to a further p50 shift related to hypercapnic respiratory failure in some critically ill COVID-19 patients, we also implemented the correction formula proposed by Dash et al. versus the Severinghaus equation. According to Equations (9a–d) and (10) in Dash et al. [4], in combination with the Hill equation with the proposed variable coefficient, we obtained the final formula:$$p50_{\mathrm{corrected}}=p50\times \frac{p50,\Delta \mathrm{pH}}{p50}\times \frac{p50,{\Delta \mathrm{CO}}_{2}}{p50}\times \frac{p50,\Delta T}{p50}$$
where
$$p50,\Delta \mathrm{pH}=p50-25.535\times \left(\mathrm{pH}-7.4\right)+10.646\times {\left(\mathrm{pH}-7.4\right)}^{2}-1.764\times {\left(\mathrm{pH}-7.4\right)}^{3}\;p50,{\Delta \mathrm{CO}}_{2}=p50+1.273\times 10^{-1}\times \left({\mathrm{pCO}}_{2}-40\right)+1.083\times 10^{-4}\times {\left({\mathrm{pCO}}_{2}-40\right)}^{2}\;p50,\Delta T=p50+1.435\times \left(T-37\right)+4.163\times 10^{-2}\times {\left(T-37\right)}^{2}+6.86\times 10^{-4}\times {\left(T-37\right)}^{3}$$

As explained in the original paper [3], due to the retrospective nature of the study, in vivo measurements of 2,3-BPG were not performed, so this correction in the formula was not used; we have therefore set it as a fixed value at 4.65 mmol/L in all arterial blood gas analyzed. Despite these correction factors, we found significant p50 temporal changes in critically ill COVID-19 patients with negative outcome, which we have analyzed and reported as an interesting mechanism never previously analyzed in critically ill COVID-19 patients (late p50s 23.0 ± 1.6 mmHg in alive patients vs. 33.1 ± 7.9 mmHg in dead patients, *p* < 0.01).

Thanks to Böning et al., who have been skeptical about low calculated p50, we were able to detect a systematic error in our calculations due to a software dysfunction: in the equation used (see above), the pCO_2_ variable resulted as zero (“0”), in general lowering all p50s. We recalculated all p50s accordingly, obtaining normal values without temporal variations comparing early p50s with late p50s, considering all analyses (26.1 ± 4.7 mmHg vs. 26.2 ± 7.0 mmHg, *p* = 0.96). However, comparing p50 temporal changes between alive and dead patients, we observed a significant difference between late p50s, 23.0 ± 1.6 mmHg in alive patients vs. 33.2 ± 7.9 mmHg in deceased patients (*p* < 0.01). As we considered a correction formula taking pCO_2_ into account, this could explain the difference between our results, and the results of Vogel et al. with a left shift of the ODC. In each case, the difference in late p50s between alive and dead patients is even more important, possibly suggesting a failure of compensation in patients with negative outcome. In this contest, we will prepare a “published erratum” in order to publish the corrected individual numerical values.

Concerning the explanation of the ODC shift, Harutyunyan et al. [5] and Wenzhong et al. [6] considered the option of a direct action of some COVID-19 proteins on heme as a certain fact; some others studies suggest different viral load concentrations during the hospital stay for COVID-19 [7,8]. Although a correlation between a change in COVID-19 viral load and a change in the ODC affinity appears plausible, we did not measured viral load, so we could not verify this hypothesis. As asked by Böning et al., none of the patients was on epoprostenol treatment (as, at the time of the first wave, this treatment for COVID-19 patients was not generally contemplated). Because we focused on the observation of p50 temporal changes in critical ill COVID-19 patients, we cannot speculate on the causes of these changes (especially in patients with negative outcome), which are remaining actually unknown. Further studies are needed to focus on this specific question in the future, to better understand pathophysiological mechanisms in the ODC affinity changes in critically ill COVID-19 patients.

## Data Availability

The data presented in this study are available on request from the corresponding author. The data are not publicly available due to privacy and ethical restrictions.

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
