# Peer review of "Reply to Böning et al. Comment on “Ceruti et al. Temporal Changes in the Oxyhemoglobin Dissociation Curve of Critically Ill COVID-19 Patients. J. Clin. Med. 2022, 11, 788”"

_jcm, 2022, doi:10.3390/jcm11154547_

Round 1

Reviewer 1 Report

I reviewed this response to Böning et al.

Are the authors stating that there was an error in the initial publication (5mmHg for each value due to a software dysfunction)? If so, this would still confirm a left shift (if I am correct 25,6mmHg for early, and 23,7mmHg for late) and that is almost exactly similar to the one Vogel et al reported  (23,4mmHg - and this could be considered „late“ given that patients are often admitted to ICU later in their disease process). May I suggest that (1) these correct values be reported, (2) an honest statement should indicate that this error was only detected because Böning et al were so sceptical, and (3) the original paper should be followed by an erratum.

The authors state: „Hopefully, we may confirm that the same statistical analysis with similar significance between p50s temporal changes between alive and death patients was observed after new recalculations;“ Does this imply these calculations have not been performed yet?

The authors did not respond to the question regarding Epoprostenol.

The authors state that their equation corrects or normalises for 2,3 BPG. Did they measure 2,3 BPG? 

Author Response

Please read the Author's notes to Reviewer into the attached files.

Round 2

Reviewer 1 Report

Thank you for the response. It is now very difficult to review because the adjusted version highlights the whole text in red not just the changes as usually required.

A few issues are still not quite clear to me. 

Firstly, the left shift:

With the new results (26.1mmHg) there still appears to be a left shift. However, I can't find statistics comparing this to the standard p50 (26.7). Has this been done?

In line with what reported by Vogel et al [2], both the English and the Swiss group reported a clear shift in Oxygen-Hemoglobin Dissociation (ODC) curve affinity, measured throughout a p50 ODC curve analysis; the English group reported the global shift in ODC p50 in critically ill COVID-19 patients compared to patients admitted to the ICU before of COVID-19 era [2], while our paper focused on p50 temporal changes during the ICU stay in critically ill COVID-19 patients [3].

It appears that the authors here reference themselves as the Swiss group and Vogel et al as the English group. This may be a bit confusing to readers who are not aware of these authors place of work. Furthermore, if Vogel et al is referred to as "the English group" the syntax doesn't make sense as it actually just says "Vogel's findings are in line with Vogel's findings and our findings".

May I suggest to state instead just simply "our findings (insert reference of own work) are in keeping with the left shift reported by Vogel et al".

The temporal trajectory:

First the authors state:

comparing early p50s with late p50s, considering all analyses (26.1 ± 4.7 mmHg vs 26.2 ± 7.0 mmHg, p = 0.96).

but then:

Despite these correction factors, we found a significant p50 temporal changes in critically ill COVID-19 patients, which we have analyzed and reported as an interesting mechanism never previously analyzed in critically ill COVID-19 patients.

I can not follow this conclusion that a p=0.96 should be significant.

CO2:

Because we considered a correction formula taking pCOinto account, this could explained the difference between our results, and the results of Vogel et al with a left shift of the ODC. 

Vogel et al corrected for CO2 in their paper.

Minor issues:

some of the sentences are very difficult to read or have grammar errors and could require some copy editing e.g.:

We also implemented the correction formula proposed by Dash et al (versus the Severinghaus equation), because of the parameters regarding especially the pCO2, which would have been eventually elevated in COVID-19 patients presenting with hypercapnic respiratory failure shifting further the curve.

Despite these correction factors, we found a significant p50 temporal changes in critically ill COVID-19 patients, which we have analyzed and reported as an interesting mechanism never previously analyzed in critically ill COVID-19 patients.

We recalculated all p50s accordingly, obtaining normal values without a temporal variations comparing early p50s with late p50s, considering all analyses (26.1 ± 4.7 mmHg vs 26.2 ± 7.0 mmHg, p = 0.96). 

ecause we considered a correction formula taking pCOinto account, this could explained the difference between our results, and the results of Vogel et al with a left shift of the ODC. 

Author Response

It is now very difficult to review because the adjusted version highlights the whole text in red not just the changes as usually required. A few issues are still not quite clear to me.  Firstly, the left shift: With the new results (26.1mmHg) there still appears to be a left shift. However, I can't find statistics comparing this to the standard p50 (26.7). Has this been done?

Response:

We thank the reviewer for these suggestions. One sample t test for early p50 (26.1 ± 4.7 mmHg, n=461) compared to 26.7 mmHg resulted statistically significant (p < 0.01), but not for late p50 (26.2 ± 7.0 mmHg, n=444, p = 0.13). This considering a hypothetical mean of 26.7 mmHg according to Severinghaus data. To clarify, we reported these values for early and late p50s in the letter.

2. In line with what reported by Vogel et al [2], both the English and the Swiss group reported a clear shift in Oxygen-Hemoglobin Dissociation (ODC) curve affinity, measured throughout a p50 ODC curve analysis; the English group reported the global shift in ODC p50 in critically ill COVID-19 patients compared to patients admitted to the ICU before of COVID-19 era [2], while our paper focused on p50 temporal changes during the ICU stay in critically ill COVID-19 patients [3]. It appears that the authors here reference themselves as the Swiss group and Vogel et al as the English group. This may be a bit confusing to readers who are not aware of these authors place of work. Furthermore, if Vogel et al is referred to as "the English group" the syntax doesn't make sense as it actually just says "Vogel's findings are in line with Vogel's findings and our findings". May I suggest to state instead just simply "our findings (insert reference of own work) are in keeping with the left shift reported by Vogel et al".

Response:

We thank the reviewer for this comment. We changed the first paragraph accordingly.

3. The temporal trajectory. First the authors state: “comparing early p50s with late p50s, considering all analyses (26.1 ± 4.7 mmHg vs 26.2 ± 7.0 mmHg, p = 0.96).” but then: “Despite these correction factors, we found a significant p50 temporal changes in critically ill COVID-19 patients, which we have analyzed and reported as an interesting mechanism never previously analyzed in critically ill COVID-19 patients”. I can not follow this conclusion that a p=0.96 should be significant.

Response:

We thank the reviewer about this question. We wanted to underline the difference between late p50 in alive vs death patients. We specified this and added the corresponding values.

4. CO2: “Because we considered a correction formula taking pCO2 into account, this could explained the difference between our results, and the results of Vogel et al with a left shift of the ODC”. Vogel et al corrected for CO2 in their paper.

Response:

We thank the reviewer for this comment. Vogel et al used the Hill equation with the corrections of Severinghaus. This formula corrected for pH, temperature and base excess, but not for CO2. In contrast, in our formula, a specific correction for pCO2 has been implemented.

5. Some of the sentences are very difficult to read or have grammar errors and could require some copy editing:

Response:

We thank the reviewer for this comment. We reviewed the manuscript and we hope it is now better.